# How Did the COVID-19 Pandemic Affect the Use of Emergency Medical Services by Patients Experiencing Mental Health Crises?

**DOI:** 10.3390/healthcare10040716

**Published:** 2022-04-13

**Authors:** Song-Yi Park, Sun-Hyu Kim

**Affiliations:** 1Department of Emergency Medicine, Dong-A University College of Medicine, Dong-A University Hospital, Busan 48114, Korea; capesong@naver.com; 2Department of Emergency Medicine, University of Ulsan College of Medicine, Ulsan University Hospital, Ulsan 44033, Korea

**Keywords:** COVID-19, mental health, emergency medical service, physical distancing

## Abstract

The COVID-19 pandemic and its resulting social restrictions have significant implications for mental health. The objective of this study was to determine the monthly trends and types of patients experiencing a mental health crisis (MHC) who used emergency medical services (EMSs) before and during the COVID-19 pandemic. A retrospective observational study was conducted using EMS data. During the study period, 8577 patients used EMSs for MHCs. EMS dispatches for MHCs and suicide completion after the COVID-19 pandemic were decreased by 12.4% and 12.7%, respectively, compared to those before the COVID-19 pandemic. Segmented regression analysis found that the number of patients per month was 6.79 before the COVID-19 pandemic. The number decreased to 4.52 patients per month during the COVID-19 pandemic, although the decrease was not statistically significant. The monthly number of patients experiencing an MHC decreased during strict social distancing measures but increased during relaxed social distancing measures. The percentage of hanging increased from 14.20% before the COVID-19 pandemic to 14.30% (*p* = 0.03) during the COVID-19 pandemic, whereas the percentages of jumping (from 15.55% to 15.28%, *p* = 0.01) and self-harm by smoke (from 4.59% to 3.84%, *p* < 0.001) during the COVID-19 pandemic were decreased compared to those before COVID-19. However, the effect size for the above findings was small (below 0.20). More than 25% of the patients experiencing an MHC who used EMSs refused to transfer to the ED over both study periods (26.49% in the pre-COVID-19 period and 28.53% in the COVID-19 period). The COVID-19 pandemic and social restrictions seemed to have some effects on the use of EMSs by MHC patients. Hanging is mainly performed indoors and is not found easily if social distancing persists, and a patient experiencing an MHC who refuses to be transferred could potentially attempt suicide. Subsequent studies should be performed to determine whether these findings are temporary during the COVID-19 pandemic or whether they will show different aspects after the COVID-19 pandemic.

## 1. Introduction

It has been almost two years since the World Health Organization declared the COVID-19 outbreak a pandemic. To control the transmission of the virus, social restrictions were imposed. These restrictions have resulted in widespread social disruptions and economic downturns [1]. They also have significant implications for mental health [2]. A recent review of the mental health outcomes of people experiencing quarantine and similar prevention measures has revealed that depression, anxiety disorders, mood disorders, posttraumatic stress disorders, sleep disorders, panic attacks, low self-esteem, and lack of self-control are prevalent in people affected by physical isolation [3].

Previous studies have reported the impact of the COVID-19 pandemic on the mental health of the general public, people who have or had COVID-19, people with pre-existing mental health disorders, and health care workers [3,4]. High federal government debt, high numbers of COVID-19 cases, low income, and the prevalence of loneliness have been found to be associated with increased long-term mental health problems in the general population in Germany [5]. COVID-19 exposure has been found to have a significant and positive relationship with nurses’ psychological distress and anxiety levels [6].

Some mental health states can be expressed in acute ways, such as self-harm, substance abuse, and suicidality; these are referred to as mental health crises (MHCs) in this study. The emergency department (ED) plays an essential role in an acute health care system; it can be a window into a public health crisis. Camilla et al. [7] found that the proportion of patients accessing the ED for suicidality was significantly higher in 2020 than in 2019. In contrast, Manuel et al. [8] reported a 52.2% decrease in psychiatric ED visits during the COVID-19 emergency state compared to the pre-COVID-19 period.

However, studies only including patients visiting EDs are insufficient. Daniel et al. [9] assessed the impact of the COVID-19 outbreak on trends in suicide-related ED visits in Madrid, Spain, and found that all-cause psychiatric ED visits have decreased. This result has limitations because it indicates that either the absolute incidence of suicide-related visits has decreased or patients are not presenting to the ED (the absolute incidence of suicide-related visits might be unchanged or increased). This limitation can be addressed by analyzing emergency medical service (EMS) data. EMSs are accessible in many countries. Patients and individuals living nearby can activate the service, making it more sensitive to discovering patients experiencing MHCs. In addition, EMS data include patients who refuse to be transported. Therefore, the aim of this study was to examine and compare the monthly trends and types of patients experiencing MHCs who used EMSs before and during the COVID-19 pandemic.

## 2. Materials and Methods

### 2.1. Study Design

This retrospective observational study investigated and compared the incidence and types of patients who used EMSs for MHCs in Busan, South Korea, before and during the COVID-19 pandemic. We defined the period before the COVID-19 pandemic as 1 March 2019, to 28 February 2020, and during the COVID-19 pandemic as 1 March 2020, to 28 February 2021. The end of February was chosen as the breakpoint because the first COVID-19 case in Busan was confirmed on 21 February 2020.

### 2.2. Study Setting

We conducted this study in Busan, which is located on the southwest coast of Korea. This 765.94 km^2^ area consists of 16 districts (one gun and 15 gu), with 3.35 million people. The EMS system in Busan is government-based and single-tiered. It provides basic to intermediate levels of EMSs from fire agency headquarters [10]. EMSs are provided free of charge to everyone as part of the country’s social security and welfare policy. Psychiatrists in five university hospitals in the region and one university hospital at the regional border are on duty at night. Emergency physicians treat intoxicated patients 24 h a day.

Social distancing measures began in Busan on 9 August 2020, with five levels (level 1, level 1.5, level 2, level 2.5, and level 3), of which level 3 was the strictest distancing measure. Before August, Busan followed a national social distancing policy. From March to 5 May 2020, stringent social distancing measures allowed everyone to stay at home as much as possible. Starting on 6 May 2020, the social distancing policy changed to distancing in daily life, loosening these measures to maintain essential everyday life activities. From 9 August to 22 August 2020, the Busan Metropolitan Government strengthened its distancing measures to level 3 due to the region’s local increase in confirmed COVID-19 cases. Then, level 2 was maintained from 23 August to 5 November and was mitigated to level 1 from 6 November to 13 December 2020. However, due to another increase in COVID-19 cases, social distancing measures were strengthened to level 2.5 from 14 December 2020, to 23 January 2021, followed by level 2 from 24 January 2021, to 14 February 2021, and level 1.5 after 14 February 2021.

### 2.3. Study Population

The inclusion criteria were all patients who used EMSs for MHCs during the study period. We did not set exclusion criteria for age to investigate the trend of EMS use in patients experiencing MHCs who belonged to the young age group in the region. We defined an MHC as any situation where one’s emotions and behaviors could result in injury to oneself or others or not performing personal care. These included the entire spectrum of suicidal thoughts, suicide attempts, and suicide completion. Specifically, we searched for and included patients who accessed EMSs themselves or with the help of others. We extracted the events based on the International Classification of Disease (ICD)-10 version 2016 as follows: (1) self-poisoning by drugs (prescribed or over-the-counter drugs; code X60-64); (2) self-poisoning by other means (organic solvents, gases, pesticides, noxious chemicals; code X66-69); (3) intentional self-harm by hanging, strangulation, or suffocation (code X70); (4) intentional self-harm by drowning or submersion (code X71); (5) intentional self-harm by smoke, fire, or flames (code X76); (6) intentional self-harm by a sharp object (code X78); (7) intentional self-harm by a blunt object (code X79); (8) intentional self-harm by jumping from a high place (code X80); (9) intentional self-harm by jumping or lying in front of a moving object (code X81); (10) intentional self-harm by crashing a motor vehicle (code X82); (11) intentional self-harm by other specified means (code X83); and (12) intentional self-harm by unspecified means (a patient who had a suicidal intention, but a specific method was not performed or a patient who left a suicide note, but the specific method of suicide was not known; code X84). We also included patients who used EMSs for anxiety, panic attacks, and depressive moods. However, suicide by firearm was not included in the classification criteria. This is because the general public is not allowed to have firearms in South Korea. Suicide by gun is extremely rare except in the military.

We did not include patients who were drunk without any additional information related to MHCs because they were indistinguishable from patients who presented with intentional self-poisoning by alcohol (code X65) and those who presented with a mental/behavioral disorder due to alcohol use (code F10) in the prehospital stage. We did not include patients with hyperventilation without information on MHCs because they could not be distinguished from those who presented with organic hyperventilation or psychogenic hyperventilation.

### 2.4. Study Outcomes

The primary outcomes were the monthly trends of patients experiencing MHCs who used EMSs. The secondary outcomes were the types of patients experiencing MHCs who used EMSs before and during the COVID-19 pandemic.

### 2.5. Data Sources and Collection

We collected anonymous emergency dispatch records from the Busan fire agency. Prehospital emergency dispatch records of all EMS dispatchers are electronically compiled from scene-dispatched EMS providers and managed by regional fire agencies. We extracted the following variables from the records: patient age and sex, EMS contact date, chief complaints resulting in EMS activation, details on present illnesses for identifying MHC patterns, and whether the patient was transferred to the ED. MHC patterns were classified into the following six categories by referring to previous studies: (1) hanging (including suffocation); (2) jumping (including drowning); (3) self-poisoning by drugs; (4) self-poisoning by substances (except drugs, including gases, pesticides, and caustics); (5) self-harm by smoke (including fire); (6) self-harm by a sharp object; (7) anxiety, panic attacks, and depressive mood; and (8) others (suicidal intention confirmed, but a specific method was not performed or identified). If the patient had two or more types of MHCs, they were classified according to the more serious medical issue. [11,12] For instance, a patient with a superficial wrist laceration and pesticide self-poisoning was classified as self-poisoning by substances.

### 2.6. Statistical Analysis

A descriptive analysis was performed to examine the study population. Continuous variables are presented as the mean and standard deviation (SD). Categorical variables are presented as numbers and proportions. When comparing the periods before and after the COVID-19 pandemic, the Kolmogorov–Smirnov test was conducted for normality. After that, we performed Student’s *t*-test and one-way analysis of variance for continuous variables and a chi-squared test for categorical variables. We also analyzed the standardized mean difference (d) with a 95% confidence interval for effect size. The reference of the effect size was based on the d value as initially suggested by Cohen and expanded by Sawilowsky: d(0.01) = very small; d(0.20) = small; d(0.50) = medium; d(0.80) = large; d(1.20) = very large; and d(2.00) = huge [13]. Pearson’s correlation analysis was performed to examine the correlation between the monthly number of patients experiencing MHCs who used EMSs and suicide completion. Segmented regression analysis was performed to compare trends in the number of monthly patients experiencing MHCs who used EMSs between the two periods (before and during the COVID-19 pandemic). All statistical analyses were performed using SPSS software (version 26.0, SPSS Inc., Chicago, IL, USA). A two-sided *p* value < 0.05 was considered statistically significant.

## 3. Results

There were 359,496 EMS dispatches during the study period (186,986 before the COVID-19 pandemic and 172,510 during the COVID-19 pandemic). We excluded 69,866 nonrescue dispatches, such as those involving public support, vehicle maintenance, education, missing records, or cancellations. Finally, 289,630 dispatches (152,747 before the COVID-19 pandemic and 136,886 during the COVID-19 pandemic) were analyzed. Among them, the data of 8577 patients were extracted according to the inclusion criteria (4571 patients before the COVID-19 pandemic and 4006 patients during the COVID-19 pandemic, Figure 1).

The mean age of the pre-COVID-19 group was 46.51 ± 18.13 years, which was slightly older than that of the COVID-19 group (45.44 ± 18.63 years). The proportion of males was 54.3% in the pre-COVID-19 group, which was higher than that (45.44%) in the COVID-19 group. There were 95 patients under 18 years old in the pre-COVID-19 group and 65 in the COVID-19 group. In the pre-COVID-19 group, the youngest patient was ten years old, and the oldest was 99 years old. In the COVID-19 group, the youngest patient was nine years old, and the oldest was 98 years old.

### 3.1. Monthly Trends in Patients Experiencing MHCs Who Used EMSs before and during the COVID-19 Pandemic

The monthly trends in MHCs are shown in Figure 2. There was a moderate correlation between the monthly number of patients experiencing MHCs and suicide completion (r-value = 0.58). Before the COVID-19 pandemic, the monthly number of patients experiencing MHCs who used EMSs steadily decreased over the year.

Segmented regression analysis found that during the study period before the COVID-19 pandemic, the number of patients decreased by 6.80 per month (*p* = 0.02). The number of patients decreased from 6.80 each month before the COVID-19 pandemic to 4.53 during the COVID-19 pandemic. However, this change was not statistically significant (*p* = 0.57) (Table 1, Figure 3). The monthly number of patients experiencing MHCs was decreased during strict social distancing measures but increased during relaxed social distancing measures (Figure 3).

### 3.2. Types of Patients Experiencing MHCs Who Used EMSs before and during the COVID-19 Pandemic

The types of patients experiencing MHCs who used EMSs are shown in Table 2. During the COVID-19 pandemic, EMS dispatches for MHCs decreased by 12.4% (4571 versus 4006, *p* < 0.001). Suicide completion also decreased by 12.7% (608 versus 531, *p* = 0.023). The most common type of MHC causing patients to use EMSs was self-poisoning by drugs in both time periods (before and during the COVID-19 pandemic). During the study period (before and during the COVID-19 pandemic), hanging increased from 14.20% to 14.30% (*p* = 0.03), whereas jumping and self-harm by smoke decreased from 15.55% to 15.28% (*p* = 0.01) and from 4.59% to 3.84% (*p* < 0.001), respectively. However, the effect size for the above findings was small (below 0.20). The most common and lethal type of suicide was hanging in both time periods (before and during the COVID-19 pandemic).

## 4. Discussion

This study attempted to identify and compare the monthly trends and types of patients experiencing MHCs who used EMSs before and during the COVID-19 pandemic in Busan, South Korea. During the COVID-19 pandemic, patients experiencing MHCs who used EMSs decreased by 12.4%. Suicide completions also decreased by 12.7%. However, the monthly trends fluctuated according to the social distancing measures. The number of patients experiencing MHCs who used EMSs decreased when social distancing was strengthened but increased when social distancing was relaxed. Regarding the types of MHCs, the proportion of hangings, which had the highest mortality rate, increased (from 14.20% to 14.30%, *p* value = 0.03), whereas that for jumping (from 15.55% to 15.28%, *p* value = 0.01) and self-harm by smoke (from 4.59% to 3.84%, *p* value < 0.001) decreased. The ratio of patients experiencing MHCs who refused transportation increased (from 26.49% to 28.53%, *p* value = 0.16), although this increase was insignificant. Other studies have also analyzed patients with mental illnesses visiting EDs or primary clinics [14]. However, the present study examined patients experiencing MHCs who used EMSs, including those who refused to be transferred to the ED. Our study had strengths in identifying patients experiencing MHCs who do not present to the ED.

Several previous studies have reported decreases in suicide completions during the COVID-19 pandemic. Jane et al. analyzed suicide trends in 21 countries in the early months of the COVID-19 pandemic and reported that the number of suicides was mainly unchanged or decreased compared to expected pre-pandemic levels [15]. Their study revealed that the rate ratio [95% confidence interval] was 0.94 [0.92–0.97] in South Korea. According to this finding, the number of suicide completions after the COVID-19 pandemic in Busan was expected to be 572 [559–590]. However, the observed number was 531, showing a more significant decrease than predicted. This might be because patients who expired after being transferred to the ED were not included in the results of the present study.

We found that the number of patients experiencing MHCs who used EMSs decreased after March 2020, which was the beginning of the pandemic. It began to increase as social distancing measures were relaxed in May 2020. These findings are consistent with those from a study by Kristin et al. [16] showing that weekly ED visit counts for mental health conditions, suicide attempts, overdoses, and violence decreased between 8 and 28 March 2020, but increased as the stay-at-home order was relaxed. If social isolation and economic difficulties persist by strengthening social distancing, they will increase anxiety and stress, exacerbating mental illnesses. Thus, patients experiencing MHCs who use EMSs will increase. However, our results showed the opposite. Two explanations could be considered for our findings. First, people might have been reluctant to visit EDs due to strong social distancing measures and the fear of infection. Second, social distancing might have a positive effect on some populations. Some people might have reduced their daily stress by staying at home. Some people might have spent more time with each other, thus strengthening their relationships. The second explanatory hypothesis requires further research to support it. The reason we deduced this hypothesis was that the mean age of the study population was the mid-40s, which is the most socially active period.

There was a reduction in the number of patients experiencing MHCs who used EMSs compared to the predicted value in the first level 3 and third level 2.5 social distancing periods. However, this was not the case in the second level 3 social distancing period (Figure 3). This seemed to be because the first and third lockdowns were implemented nationwide, but the second lockdown was briefly limited to Busan. The scale of the regulation might have also affected the use of EMSs by patients experiencing MHCs.

We found that the rate of hanging in cases of MHCs increased during the COVID-19 pandemic, although the effect size was very small (d = 0.12). Graham et al. reported an increase in weekly hanging cases, with the median value doubling after a lockdown in the United Kingdom [17]. Hanging is mainly performed indoors and may not be easily found if social distancing persists, unlike jumping or self-harm by smoke. We were able to find quite a few records of suicide by hanging that were discovered after a long period of time. This phenomenon (kodokushi or lonely death) is expected to increase further if social distancing without social connections continues [18].

Since the start of the COVID-19 pandemic, self-poisoning by drugs (drug overdoses) seems to be increasing significantly worldwide [19,20]. Svetla et al. [21] reported a 17% increase in opioid overdose EMS transportation to the ED and a 50% increase in responses to suspected opioid overdoses with death at the scene. Nancy et al. [22] found that drug overdose EMS calls and deaths from drug overdoses were increased by 43% and 47%, respectively, compared to those before the COVID-19 pandemic. In our study, the rate of self-poisoning by drugs increased from 22.73% to 25.71% during the COVID-19 pandemic, although the increase was insignificant. However, the mortality from self-poisoning was less than 1% in both time periods (before and during the COVID-19 pandemic). Opioid overdose is not frequent in South Korea [23]. This is because opioid prescriptions are controlled as part of an opioid abuse prevention program embedded in the Narcotics Information Management System [24]. Hence, the drugs involved in overdoses are not opioids but are usually psychoactive or over-the-counter medications such as painkillers [25]. This difference in drug types seems to be associated with a low mortality rate in drug overdoses.

More than a quarter of patients experiencing MHCs refused to be transferred to the ED in both study periods (26.49% in the pre-COVID-19 period and 28.53% in the COVID-19 period). Patients might refuse to be transferred for many reasons, such as the fear of contagion, concern over costs, and the fear of being a burden to their families. In a study conducted in Israel, being female and having social vulnerability were significantly associated with higher transport refusal [26]. Whatever the reason for refusal was, some patients who refused transportation attempted suicide. The risk of suicide after a suicide attempt persists for a long time [27]. Individuals who complete suicide are less likely to be followed by mental health services than those who attempt suicide [28]. Urgent interventions and follow-up research on patients experiencing MHCs who refuse transport are needed.

This study has several limitations. First, the EMS system varies by country. Some patients might not be able to use EMSs. In some cases of suicide where the cause of death is unclear, administrative procedures may proceed without contacting EMSs. These cases would not have been included in this study. Thus, it may not be reasonable to generalize our results. Second, suicide completion was represented only by patients who completed suicide attempts and whose resuscitation was withheld by EMS providers because death was apparent at the scene. Patients who were transported to the ED under resuscitation were classified as the MHC group. However, we estimated that the number of these patients was small. Third, it was impossible to distinguish whether an event was a suicide or an accident using records in the case of drownings. Fortunately, there were few cases of this. Fourth, the effect size for the trend change in the MHC type before and after the COVID-19 pandemic was not large. However, this was the result of the first-year analysis of the COVID-19 pandemic. Further follow-up studies are needed. Fifth, the same patients might have used EMSs several times. Data received from the fire agency did not contain personal information to confirm this. Even if the same patient had used EMSs several times, if there were changes before and after the COVID-19 pandemic, it would suggest that the lockdown affected their EMS utilization. Finally, we failed to analyze age and sex associated with MHCs because there were many missing data for patients who refused transportation to the ED.

## 5. Conclusions

In conclusion, during the COVID-19 pandemic, patients experiencing MHCs who used EMSs and suicide completion decreased by approximately 12%, especially when social distancing measures were strengthened, showing a tendency to decrease further. Among the types of MHCs, the proportion of hanging slightly increased, whereas the proportions of jumping and self-harm by smoke decreased. More than 25% of patients experiencing MHCs who used EMSs refused to be transferred to the ED over the study period. Hanging is mainly performed indoors and is not found easily if social distancing persists, and a patient experiencing an MHC who refuses to be transferred could potentially attempt suicide. Subsequent studies should be performed to determine whether these findings are temporary during the COVID-19 pandemic or whether they will show different aspects after the COVID-19 pandemic. Quarantine authorities need to emphasize that social interaction should continue as much as social distancing.

## Figures and Tables

**Figure 1 healthcare-10-00716-f001:**
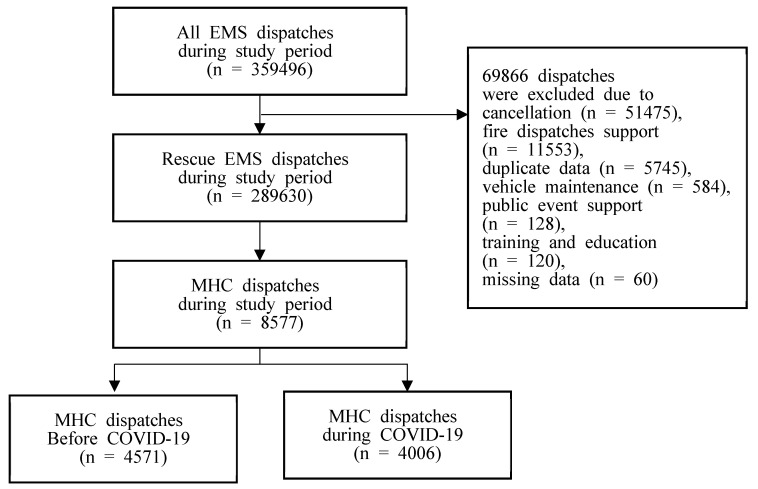
Flow chart of the study population.EMS, emergency medical service; MHC, mental health crisis; COVID-19, coronavirus disease-2019.

**Figure 2 healthcare-10-00716-f002:**
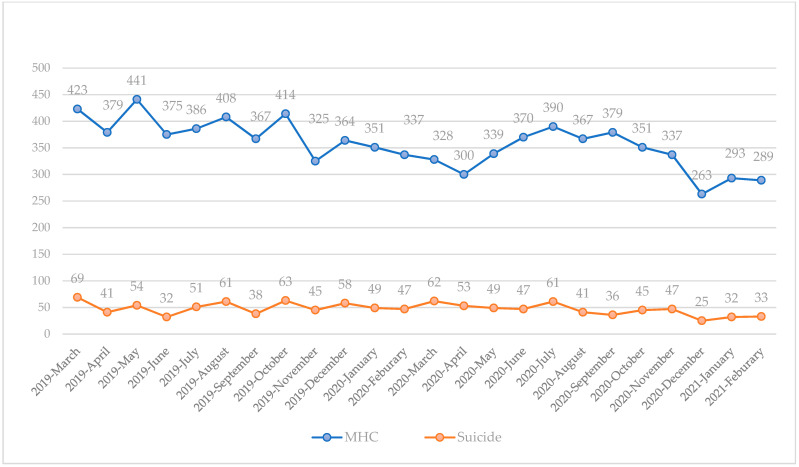
Monthly trends in patients experiencing MHCs who used EMSs before and during the COVID-19 pandemic. MHC, mental health crisis; EMS, emergency medical service. The *Y*-axis on the left is the number of patients experiencing MHCs who used EMSs.

**Figure 3 healthcare-10-00716-f003:**
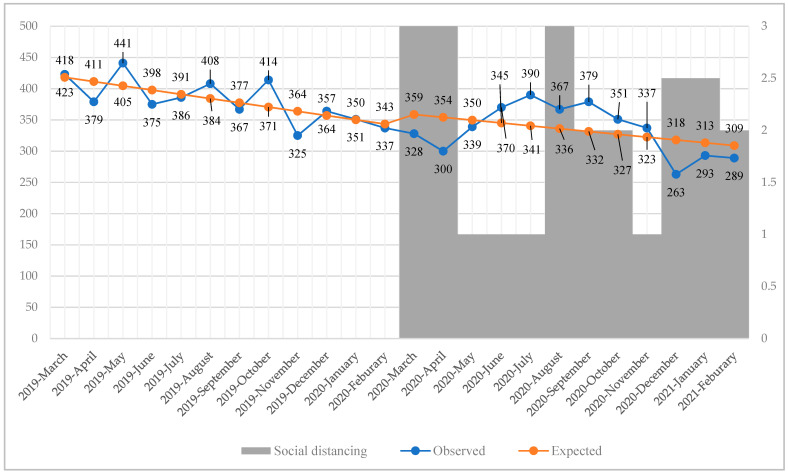
Comparison of the observed monthly trends in patients experiencing MHCs who used EMSs and the numbers expected by segmented regression analysis. The *Y*-axis on the left is the number of patients experiencing MHCs who used EMSs. The right *Y*-axis is the social distancing level (level 3 was the strictest). The blue dotted line is the observed number of patients. The orange dotted line was derived by segmented regression analysis. The breakpoint was March 2020. MHC, mental health crisis; EMS, emergency medical service.

**Table 1 healthcare-10-00716-t001:** Parameter estimates, standard errors, and *p* values from the segmented regression models predicting the mean monthly number of patients experiencing MHCs who used EMSs.

	Coefficient	Standard Error	t-Statistic	*p* Value
Pre-COVID-19 intercept	425.02	20.40	20.83	<0.00
Pre-COVID-19 slope	−6.80	2.77	−2.45	0.02
COVID-19 intercept change	19.79	27.21	0.73	0.48
COVID-19 slope change	2.27	3.92	0.58	0.57

MHC, Mental health crisis; EMS, emergency medical service; COVID-19, coronavirus disease-2019.

**Table 2 healthcare-10-00716-t002:** Comparison of the types of patients experiencing MHCs who used EMSs before and during the COVID-19 pandemic.

	Pre-COVID-19	COVID-19	*p* Value	Effect Size, d	95% CI	
**All rescue EMS dispatches**	152,747	136,883	<0.00	0.01	0.01	0.02
MHC dispatches	4571 (2.99%)	4006 (2.93%)	<0.00	0.07	0.03	0.11
Age (mean ± SD) *	46.51 ± 18.13	45.44 ± 18.63	0.01	−0.06	−0.10	−0.01
Sex (male) *	2137 (54.3%)	1800 (45.7%)	0.03	0.05	0.01	0.09
**MHC type**						
Hanging	649 (14.20%)	573 (14.30%)	0.03	0.12	0.01	0.24
Jumping	711 (15.55%)	612 (15.28%)	0.01	0.14	0.03	0.25
Self-poisoning by drugs	1039 (22.73%)	1030 (25.71%)	0.84	0.01	−0.08	0.10
Self-poisoning by substances	222 (4.86%)	205 (5.12%)	0.41	0.08	−0.11	0.27
Self-harm by smoke	210 (4.59%)	154 (3.84%)	<0.00	0.32	0.11	0.53
Self-harm by a sharp object	927 (20.28%)	889 (22.19%)	0.37	0.04	−0.05	0.13
Anxiety, panic attacks, and depressive mood	488 (10.68%)	427 (10.66%)	0.04	0.13	0.00	0.26
Others (nonidentified)	325 (7.11%)	116 (2.90%)	<0.00	0.36	0.15	0.57
**Suicide completion**	608 (53.38%)	531 (46.62%)	0.02	0.14	0.02	0.25
Hanging	391 (64.31%) (60.25%)	367 (69.11%) (64.05%)	0.38	0.06	−0.08	0.21
Jumping	106 (17.43%) (14.91%)	84 (15.82%) (13.73%)	0.11	0.23	−0.05	0.52
Self-poisoning by drugs	2 (0.33%) (0.19%)	5 (0.94%) (0.49%)	0.26	1.07	−0.66	2.80
Self-poisoning by substances	5 (0.82%) (2.25%)	5 (0.94%) (2.44%)	1.00	0.00	−1.24	1.24
Self-harm by smoke	56 (9.21%) (26.67%)	43 (8.10%) (27.92%)	0.19	0.27	−0.13	0.67
Self-harm by a sharp object	38 (6.25%) (4.10%)	22 (4.14%) (2.47%)	0.04	0.57	0.03	1.10
Anxiety, panic attacks, and depressive mood	0 (0.00%) (0.00%)	0 (0.00%) (0.00%)	-	-	-	-
Others (nonidentified)	10(1.64%) (3.08%)	5 (0.94%) (4.31%)	0.20	0.74	−0.36	1.85
**Refused transportation**	1211 (26.49%)	1143 (28.53%)	0.16	0.06	−0.02	0.14

MHC, mental health crisis; EMS, emergency medical service; COVID-19, coronavirus disease-2019; SD, standard deviation; CI, confidence interval. Variables are presented as numbers (percentages) (mortality). * There were many missing age and sex data points. This result reflects the analysis excluding the missing data.

## Data Availability

Raw data were generated by the national fire agencies in Korea. Derived data supporting the findings of this study are available from the corresponding author (S.H.K.) upon reasonable request.

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
