# Peer review of "How Did the COVID-19 Pandemic Affect the Use of Emergency Medical Services by Patients Experiencing Mental Health Crises?"

_healthcare, 2022, doi:10.3390/healthcare10040716_

Round 1

Reviewer 1 Report

Overall this is a well written, thoughtful paper on an important topic in public health research. The major limitation is generalizability given the data are from one city in Korea. The data would be bolstered by demonstrating similar trends in other areas of the country or internationally. I only have a few suggestions for the authors:

  1. Given your sample size, some of the effect sizes for significant tests are quite small. For example, Table 2 compares pre and post-COVID 19 rates of MHC. Although the comparison of hangings is statistically significant, it's not clear that the effect size (14.2% vs. 14.3%) is meaningful. The authors should provide a rationale for the public health significance of very small, albeit statistically significant comparisons.
  2. Is there a way of knowing if the pre-covid and post-covid cases are independent? It seems plausible that a significant proportion of cases in the two periods are identical people.
  3. I thought Figure 3 was the most interesting in the paper. I was struck by the jump in observed cases after the end of the lockdown. The authors provide a nice discussion of this but the mechanism of this relationship is puzzling, particularly given it did not occur following the second Level 3 lockdown. Any ideas why it occurred after one lockdown but not the other?
  4. Figure 2 suggests that, overall, both MHC and Suicide are going down over time. It looks like the pandemic didn't change the trajectory of MHC and Suicide over the long term. Any ideas why the trends are in the positive direction?

Very nice work - my kudos to the authors.

Author Response

Thank you for your valuable time and comments. These comments have improved the quality of our manuscript significantly. We have revised this manuscript according to your comments or suggestions.

Comments and Suggestions for Authors
Reviewer: 1

  1. Given your sample size, some of the effect sizes for significant tests are quite small. For example, Table 2 compares pre and post-COVID 19 rates of MHC. Although the comparison of hangings is statistically significant, it's not clear that the effect size (14.2% vs. 14.3%) is meaningful. The authors should provide a rationale for the public health significance of very small, albeit statistically significant comparisons.

Answer;

Thank you for the critical comment. We added the effect size and 95% confidence intervals to Table 2. However, the overall effect size was small. Regarding this, we added the following sentences to the limitation section.

Previous version

Table 2. Comparison of types of MHC patients using EMSs before and during COVID-19

preCOVID-19

COVID-19

P-value

All rescue EMS dispatches

152747

136883

<0.00

MHC dispatches

4571 (2.99%)

4006 (2.93%)

<0.00

Age (mean±SD)*

46.51±18.13

45.44±18.63

0.01

Gender (male)*

2137 (54.3%)

1800 (45.7%)

0.03

MHC types

Hanging

649 (14.20%)

573 (14.30%)

0.03

Jumping

711 (15.55%)

612 (15.28%)

0.01

Self-poisoning by drugs

1039 (22.73%)

1030 (25.71%)

0.84

Self-poisoning by substances

222 (4.86%)

205 (5.12%)

0.41

Self-harm by smoke

210 (4.59%)

154 (3.84%)

<0.00

Self-harm by a sharp object

927 (20.28%)

889 (22.19%)

0.37

Anxiety, panic attack, and

Depressive mood

488 (10.68%)

427 (10.66%)

0.04

Others (non-identified)

325 (7.11%)

116 (2.90%)

<0.00

Suicide completion

608 (53.38%)

531 (46.62%)

0.02

Hanging

391 (64.31%) (60.25%)

367 (69.11%) (64.05%)

0.38

Jumping

106 (17.43%) (14.91%)

84 (15.82%) (13.73%)

0.11

Self-poisoning by drugs

2 (0.33%) (0.19%)

5 (0.94%) (0.49%)

0.26

Self-poisoning by substances

5 (0.82%) (2.25%)

5 (0.94%) (2.44%)

1.00

Self-harm by smoke

56 (9.21%) (26.67%)

43 (8.10%) (27.92%)

0.19

Self-harm by a sharp object

38 (6.25%) (4.10%)

22 (4.14%) (2.47%)

0.04

Anxiety, panic attack, and

depressive mood

0 (0.00%) (0.00%)

0 (0.00%) (0.00%)

-

Others (non-identified)

10(1.64%) (3.08%)

5 (0.94%) (4.31%)

0.20

Refused transportation

1211 (26.49%)

1143 (28.53%)

0.16

MHC, mental health crisis; EMS, emergency medical service; COVID-19, coronavirus Disease-19; SD, standard deviation. Variables are presented as numbers (percentages) (mortality). *There were many missing age and gender data points. This result reflects analysis excluding the missing data.

Revised version

Table 2. Comparison of types of MHC patients using EMSs before and during COVID-19.

preCOVID-19

COVID-19

P-value

Effect size, d

95% CI

All rescue EMS dispatches

152747

136883

<0.00

0.01

0.01

0.02

MHC dispatches

4571 (2.99%)

4006 (2.93%)

<0.00

0.07

0.03

0.11

Age (mean±SD)*

46.51±18.13

45.44±18.63

0.01

-0.06

-0.10

-0.01

Gender (male)*

2137 (54.3%)

1800 (45.7%)

0.03

0.05

0.01

0.09

MHC types

Hanging

649 (14.20%)

573 (14.30%)

0.03

0.12

0.01

0.24

Jumping

711 (15.55%)

612 (15.28%)

0.01

0.14

0.03

0.25

Self-poisoning by drugs

1039 (22.73%)

1030 (25.71%)

0.84

0.01

-0.08

0.10

Self-poisoning by substances

222 (4.86%)

205 (5.12%)

0.41

0.08

-0.11

0.27

Self-harm by smoke

210 (4.59%)

154 (3.84%)

<0.00

0.32

0.11

0.53

Self-harm by a sharp object

927 (20.28%)

889 (22.19%)

0.37

0.04

-0.05

0.13

Anxiety, panic attack, and

Depressive mood

488 (10.68%)

427 (10.66%)

0.04

0.13

0.00

0.26

Others (non-identified)

325 (7.11%)

116 (2.90%)

<0.00

0.36

0.15

0.57

Suicide completion

608 (53.38%)

531 (46.62%)

0.02

0.14

0.02

0.25

Hanging

391 (64.31%) (60.25%)

367 (69.11%) (64.05%)

0.38

0.06

-0.08

0.21

Jumping

106 (17.43%) (14.91%)

84 (15.82%) (13.73%)

0.11

0.23

-0.05

0.52

Self-poisoning by drugs

2 (0.33%) (0.19%)

5 (0.94%) (0.49%)

0.26

1.07

-0.66

2.80

Self-poisoning by substances

5 (0.82%) (2.25%)

5 (0.94%) (2.44%)

1.00

0.00

-1.24

1.24

Self-harm by smoke

56 (9.21%) (26.67%)

43 (8.10%) (27.92%)

0.19

0.27

-0.13

0.67

Self-harm by a sharp object

38 (6.25%) (4.10%)

22 (4.14%) (2.47%)

0.04

0.57

0.03

1.10

Anxiety, panic attack, and

Depressive mood

0 (0.00%) (0.00%)

0 (0.00%) (0.00%)

-

-

-

-

Others (non-identified)

10(1.64%) (3.08%)

5 (0.94%) (4.31%)

0.20

0.74

-0.36

1.85

Refused transportation

1211 (26.49%)

1143 (28.53%)

0.16

0.06

-0.02

0.14

MHC, mental health crisis; EMS, emergency medical service; COVID-19, coronavirus Disease-19; SD, standard deviation; CI, confidence interval. Variables are presented as numbers (percentages) (mortality). *There were many missing age and gender data points. This result reflects analysis excluding the missing data.

Fourth, the effect size for the trend change in the MHC type before and after COVID-19 was not large. However, this was the result of the COVID-19 pandemic first-year analysis, which requires further follow-up studies.

  1. Is there a way of knowing if the pre-covid and post-covid cases are independent? It seems plausible that a significant proportion of cases in the two periods are identical people.

Answer:

That is a very reasonable point. The same patients had likely used EMSs several times. The data we received from the fire agency did not contain the user’s personal information to confirm this. However, even if the same patient had used EMSs several times, this is also meaningful as it suggests that EMS use was affected by the lockdown in the same patient. Regarding this, we added the following sentences to the limitation section.

Fifth, the same patients may have used EMSs several times. The data received from the fire agency did not contain personal information to confirm this. However, we believe that even if the same patient had used EMSs several times, if there were changes before and after COVID-19, this suggests that the lockdown affected EMS utilization.  

  1. I thought Figure 3 was the most interesting in the paper. I was struck by the jump in observed cases after the end of the lockdown. The authors provide a nice discussion of this, but the mechanism of this relationship is puzzling, particularly given it did not occur following the second Level 3 lockdown. Any ideas why it occurred after one lockdown but not the other?

Answer:

That is a good point. During the first and third lockdown, the number of MHC patients using EMS decreased more than expected, but not in the second lockdown. Our hypothesis for this finding is as follows. The first and third lockdowns were implemented in Busan and across the country due to an explosive increase in the number of confirmed patients. However, the second lockdown was a short one limited to Busan as the number of confirmed patients returning to Busan from abroad increased. The scale of the regulation is estimated to have affected the use of EMSs by MHC patients. We added the following to the Discussion section.

Interestingly, there was a relatively small reduction in the second level 3 social distancing compared to the first and third level 3 social distancing periods. This seems to be because the first and third lockdowns were implemented nationwide, but the second was limited to Busan and only briefly. The scale of the regulation is also estimated to have affected the use of EMSs by MHC patients.   

  1. Figure 2 suggests that, overall, both MHC and Suicide are going down over time. It looks like the pandemic didn't change the trajectory of MHC and Suicide over the long term. Any ideas why the trends are in the positive direction?

Answer:

Studies on suicide seasonality in Korea reported the highest suicide rate in spring and a gradual decline thereafter [1,2]. This is consistent with the results of our study shown in Figure 2. Several hypotheses can be applied to explain the findings. However, what we think is most relevant is that schools, companies, and public institutions in Korea start their work on March 1st and end on February 28th of the following year. Thus, it is estimated that March and April are the most stressful periods for beginners in Korea, so the suicide rate is the highest in spring.

However, we thought that this part was a little outside the discussion of this study, so we did not add it to the main text.

Reference

  1. Yang, C. T., Yip, P. S., Cha, E. S., & Zhang, Y. (2019). Seasonal changes in suicide in South Korea, 1991 to 2015. Plos one14(6), e0219048.
  2. Yu, J., Yang, D., Kim, Y., Hashizume, M., Gasparrini, A., Armstrong, B., ... & Chung, Y. (2020). Seasonality of suicide: a multi-country multi-community observational study. Epidemiology and psychiatric sciences29.

Reviewer 2 Report

Dear Authors,

Thank you for the opportunity to review this important work considering the association between mental health crisis and the emergency medical services before and during the COVID-19 pandemic. I found this work interesting since it can shed new light on the impact of the pandemic. However, some major changes should be addressed before its official publication in Healthcare Journal.
All the comments are reported below.

Abstract

  • I would suggest the authors to provide few words at the beginning of the abstract to contextualize this work and then state the goal.
  • Since the “refusing transportation” result is not significant, I would be very careful to report it in the discussion part of the abstract. I can be worth discussing it in the discussion later in the paper but not sure it’s appropriate to report this result in the abstract since it’s not significant and can be misleading.
  • I would suggest the authors to draw some conclusions answering why is this study important?
  •  

Introduction

  • I suggest the authors to provide additional evidence considering mental health related to COVID 19 pandemic to provide more robust theoretical background. These references might be helpful <<Hossain, M. M., Tasnim, S., Sultana, A., Faizah, F., Mazumder, H., Zou, L., ... & Ma, P. (2020). Epidemiology of mental health problems in COVID-19: a review. F1000Research9; Moreno, C., Wykes, T., Galderisi, S., Nordentoft, M., Crossley, N., Jones, N., ... & Arango, C. (2020). How mental health care should change as a consequence of the COVID-19 pandemic. The Lancet Psychiatry, 7(9), 813-824.>> and others.

Study population

  • Age range of participants should be taken in consideration given that different pattern of data can emerge considering younger/older participants.

Statistical analysis

  • Are the variables checked for normality and linearity before the application of parametric tests? Which tests do the authors use for testing normality and linearity? Please provide more information about this is the “statistical analysis” section.

Results

  • In general, given the correlational nature of this study, interpretation of the results should be taken with caution.
  • Numeric results in brackets should be reported extensively (r value, p value…

Discussion  

  • The explanation that social distancing may have had positive effects on some populations might be strong and, for this, it needs more explanation and support. The authors should also consider and think about possible intervening variables that could have influenced these results. For example, age of the participants and gender.
  • Also, the interpretation considering refusing transportation should be taken with caution, also because the increasing rate was not significant.
  • In general, results should be interpreted also considering that some patients could not have used EMCs, and, therefore, generalization is not possible.

Author Response

Thank you for your valuable time and comments. These comments have improved the quality of our manuscript significantly. We have revised this manuscript according to your comments or suggestions.

Reviewer: 2
Dear Authors,

Thank you for the opportunity to review this important work considering the association between mental health crisis and the emergency medical services before and during the COVID-19 pandemic. I found this work interesting since it can shed new light on the impact of the pandemic. However, some major changes should be addressed before its official publication in Healthcare Journal.
All the comments are reported below.

Abstract

  • I would suggest the authors to provide few words at the beginning of the abstract to contextualize this work and then state the goal.

Answer;

Thank you for your suggestion. We added the following sentences in the abstract.

The COVID-19 pandemic and its resulting social restriction have had significant implications for mental health. 

  • Since the “refusing transportation” result is not significant, I would be very careful to report it in the discussion part of the abstract. I can be worth discussing it in the discussion later in the paper but not sure it’s appropriate to report this result in the abstract since it’s not significant and can be misleading.

Answer;

We totally agree with you. The description regarding this has been deleted and the abstract has been revised as follows.

Before

Hanging increased over both study periods, while jumping and self-harm by smoke were decreased. Although insignificant, the proportion of patients who refused transportation rose from 26.49% to 28.53%. During the COVID-19 period, MHC patients using EMSs and the incidence of suicides decreased. The monthly trends in MHC patients using EMSs were affected by the social distancing measures. The increased rates of hanging and refusing transportation are more worrisome and require consistent attention because they may not be readily apparent events.

After

Hanging increased (14.20% to 14.30%, P = 0.03) over both study periods, while jumping (15.55% to 15.28%, P = 0.01) and self-harm by smoke (4.59% to 3.84%, P < 0.001) were decreased, but their effect size were all small.

  • I would suggest the authors to draw some conclusions answering why is this study important?

Answer;

Thank you for your constructive opinion. We admit that the conclusion was insufficient. Therefore, we revised it in the abstract as follows.

The COVID-19 and social restriction seem to have some implications on the use of EMS in MHC patients. The increased rate of hanging, which had the highest mortality, is more worrisome and requires consistent attention. It may not be readily revealed because it is done indoors. Follow-up studies are essential for determining whether this trend is temporary or rebounding.

Introduction

  • I suggest the authors to provide additional evidence considering mental health related to COVID 19 pandemic to provide more robust theoretical background. These references might be helpful <<Hossain, M. M., Tasnim, S., Sultana, A., Faizah, F., Mazumder, H., Zou, L., ... & Ma, P. (2020). Epidemiology of mental health problems in COVID-19: a review. F1000Research9Moreno, C., Wykes, T., Galderisi, S., Nordentoft, M., Crossley, N., Jones, N., ... & Arango, C. (2020).

How mental health care should change as a consequence of the COVID-19 pandemic. The Lancet Psychiatry, 7(9), 813-824.>> and others.

Answer;

Thank you for your suggestion. It was excellent research that provided theoretical background in this field that we, as emergency physicians, were not familiar with. We added the following and references to the introduction section for readers.  

A recent review of mental health outcomes of quarantine and similar prevention measures revealed that depression, anxiety disorders, mood disorders, post-traumatic stress disorders, sleep disorders, panic attacks, low self-esteem, and lack of self-control are prevalent among people affected by physical isolation [3].

Studies report the impact of the COVID-19 pandemic on the mental health of the general public, people who have or had COVID-19, people with pre-existing mental health disorders, and healthcare workers [3,4].

Study population

  • Age range of participants should be taken in consideration given that different pattern of data can emerge considering younger/older participants.

Answer;

That is a good point. While some clinical studies that show the effectiveness of drugs exclude certain ages from the study population. However, we think these exclusion criteria are not appropriate in MHC studies. Thus we added the following sentence to the study population section.

We did not set age exclusion criteria, considering that participants may have a different data pattern. 

Statistical analysis

  • Are the variables checked for normality and linearity before the application of parametric tests? Which tests do the authors use for testing normality and linearity? Please provide more information about this is the “statistical analysis” section.

Answer;

Thank you for the excellent comment. We performed the Kolmogorov-Smirnov test for normality verification since (n) was over two thousand. We checked the linearity through a scatterplot. The statistical analysis part was revised and supplemented as follows.

  • Figure 1. Linearity for before COVID-19 data

  • Figure 1. Linearity for after COVID-19 data

Before

The characteristics of the patients before and during the COVID-19 period were compared using a t-test for continuous variables and a chi-squared test for categorical variables. Pearson’s correlation analysis was performed to examine the correlation between the monthly numbers of MHC patients using EMSs and suicide completion. Segmented regression analysis was performed to compare the trends in the numbers of monthly MHC patients using EMSs between the two study periods. All statistical analyses were performed using SPSS software (version 26.0, SPSS inc., Chicago, IL, USA). The significance level was set at P < 0.05.  

After

We performed a descriptive analysis to examine the study population, Continuous variables were presented as the mean and standard deviation (SD), and categorical variables were presented as numbers and proportions. In comparing the before and after COVID-19 groups, we conducted the Kolmogorov-Smirnov test for normality. After that, we performed a Student t-test and one-way analysis of variance for continuous variables and a chi-squared test for categorical variables. We also conducted standardized mean difference (d) with a 95% confidence interval for effect size. The reference of the effect size was based on d as initially suggested by Cohen and expanded by Sawilowsky; d(0.01) is very small, d(0.20) is small, d(0.50) is medium, d(0.80) is large, d(1.20) is very large, and d(2.00) is huge [12]. Pearson’s correlation analysis was performed to examine the correlation between the monthly numbers of MHC patients using EMSs and suicide completion. Segmented regression analysis was performed to compare the trends in the numbers of monthly MHC patients using EMSs between the two study periods. All statistical analyses were performed using SPSS software (version 26.0, SPSS inc., Chicago, IL, USA). A two-sided P value of <0.05 was considered statistically significant.   

Results

  • In general, given the correlational nature of this study, interpretation of the results should be taken with caution.
  • Numeric results in brackets should be reported extensively (r-value, p value…

Answer;

Thank you for your constructive comment. We should have been more careful in interpreting the results. We added effect size in table 2, deleted all statistically insignificant result descriptions, and revised as follows.

Before

Table 2. Comparison of types of MHC patients using EMSs before and during COVID-19

preCOVID-19

COVID-19

P-value

All rescue EMS dispatches

152747

136883

<0.00

MHC dispatches

4571 (2.99%)

4006 (2.93%)

<0.00

Age (mean±SD)*

46.51±18.13

45.44±18.63

0.01

Gender (male)*

2137 (54.3%)

1800 (45.7%)

0.03

MHC types

Hanging

649 (14.20%)

573 (14.30%)

0.03

Jumping

711 (15.55%)

612 (15.28%)

0.01

Self-poisoning by drugs

1039 (22.73%)

1030 (25.71%)

0.84

Self-poisoning by substances

222 (4.86%)

205 (5.12%)

0.41

Self-harm by smoke

210 (4.59%)

154 (3.84%)

<0.00

Self-harm by a sharp object

927 (20.28%)

889 (22.19%)

0.37

Anxiety, panic attack, and

Depressive mood

488 (10.68%)

427 (10.66%)

0.04

Others (non-identified)

325 (7.11%)

116 (2.90%)

<0.00

Suicide completion

608 (53.38%)

531 (46.62%)

0.02

Hanging

391 (64.31%) (60.25%)

367 (69.11%) (64.05%)

0.38

Jumping

106 (17.43%) (14.91%)

84 (15.82%) (13.73%)

0.11

Self-poisoning by drugs

2 (0.33%) (0.19%)

5 (0.94%) (0.49%)

0.26

Self-poisoning by substances

5 (0.82%) (2.25%)

5 (0.94%) (2.44%)

1.00

Self-harm by smoke

56 (9.21%) (26.67%)

43 (8.10%) (27.92%)

0.19

Self-harm by a sharp object

38 (6.25%) (4.10%)

22 (4.14%) (2.47%)

0.04

Anxiety, panic attack, and

depressive mood

0 (0.00%) (0.00%)

0 (0.00%) (0.00%)

-

Others (non-identified)

10(1.64%) (3.08%)

5 (0.94%) (4.31%)

0.20

Refused transportation

1211 (26.49%)

1143 (28.53%)

0.16

MHC, mental health crisis; EMS, emergency medical service; COVID-19, coronavirus Disease-19; SD, standard deviation. Variables are presented as numbers (percentages) (mortality). *There were many missing age and gender data points. This result reflects analysis excluding the missing data.

After

Table 2. Comparison of types of MHC patients using EMSs before and during COVID-19.

preCOVID-19

COVID-19

P-value

Effect size, d

95% CI

All rescue EMS dispatches

152747

136883

<0.00

0.01

0.01

0.02

MHC dispatches

4571 (2.99%)

4006 (2.93%)

<0.00

0.07

0.03

0.11

Age (mean±SD)*

46.51±18.13

45.44±18.63

0.01

-0.06

-0.10

-0.01

Gender (male)*

2137 (54.3%)

1800 (45.7%)

0.03

0.05

0.01

0.09

MHC types

Hanging

649 (14.20%)

573 (14.30%)

0.03

0.12

0.01

0.24

Jumping

711 (15.55%)

612 (15.28%)

0.01

0.14

0.03

0.25

Self-poisoning by drugs

1039 (22.73%)

1030 (25.71%)

0.84

0.01

-0.08

0.10

Self-poisoning by substances

222 (4.86%)

205 (5.12%)

0.41

0.08

-0.11

0.27

Self-harm by smoke

210 (4.59%)

154 (3.84%)

<0.00

0.32

0.11

0.53

Self-harm by a sharp object

927 (20.28%)

889 (22.19%)

0.37

0.04

-0.05

0.13

Anxiety, panic attack, and

Depressive mood

488 (10.68%)

427 (10.66%)

0.04

0.13

0.00

0.26

Others (non-identified)

325 (7.11%)

116 (2.90%)

<0.00

0.36

0.15

0.57

Suicide completion

608 (53.38%)

531 (46.62%)

0.02

0.14

0.02

0.25

Hanging

391 (64.31%) (60.25%)

367 (69.11%) (64.05%)

0.38

0.06

-0.08

0.21

Jumping

106 (17.43%) (14.91%)

84 (15.82%) (13.73%)

0.11

0.23

-0.05

0.52

Self-poisoning by drugs

2 (0.33%) (0.19%)

5 (0.94%) (0.49%)

0.26

1.07

-0.66

2.80

Self-poisoning by substances

5 (0.82%) (2.25%)

5 (0.94%) (2.44%)

1.00

0.00

-1.24

1.24

Self-harm by smoke

56 (9.21%) (26.67%)

43 (8.10%) (27.92%)

0.19

0.27

-0.13

0.67

Self-harm by a sharp object

38 (6.25%) (4.10%)

22 (4.14%) (2.47%)

0.04

0.57

0.03

1.10

Anxiety, panic attack, and

Depressive mood

0 (0.00%) (0.00%)

0 (0.00%) (0.00%)

-

-

-

-

Others (non-identified)

10(1.64%) (3.08%)

5 (0.94%) (4.31%)

0.20

0.74

-0.36

1.85

Refused transportation

1211 (26.49%)

1143 (28.53%)

0.16

0.06

-0.02

0.14

MHC, mental health crisis; EMS, emergency medical service; COVID-19, coronavirus Disease-19; SD, standard deviation;CI, confidence interval. Variables are presented as numbers (percentages) (mortality). *There were many missing age and gender data points. This result reflects analysis excluding the missing data.

Before

The types of MHC patients using EMSs are shown in Table 2. During the COVID-19 period, the EMS dispatches for MHCs decreased by 12.4% (4571 versus 4006, P < 0.001), and suicide completion also decreased by 12.7% (608 versus 531, P = 0.023). Although insignificant, the proportion of patients who refused transportation increased from 26.49% to 28.53% (P = 0.16). The most common type of MHC causing patients to use EMSs was self-poisoning by drugs in both study periods. Hanging increased across both study periods from 14.20% to 14.30% (P = 0.03), whereas jumping and self-harm by smoke were decreased. The most common and lethal type of suicide was hanging in both study periods. The proportion of hanging increased within the COVID-19 study period but was insignificant (P = 0.38).

After

The types of MHC patients using EMSs are shown in Table 2. During the COVID-19 period, the EMS dispatches for MHCs decreased by 12.4% (4571 versus 4006, P < 0.001), and suicide completion also decreased by 12.7% (608 versus 531, P = 0.023). The most common type of MHC causing patients to use EMSs was self-poisoning by drugs in both study periods. Hanging increased across both study periods from 14.20% to 14.30% (P = 0.03), whereas jumping (from 15.55% to 15.28%, P = 0.01) and self-harm by smoke (from 4.59% to 3.84%, P  < 0.001) were decreased. However, the effect size for above was small, below 0.20. The most common and lethal type of suicide was hanging in both study periods. 

Discussion  

  • The explanation that social distancing may have had positive effects on some populations might be strong and, for this, it needs more explanation and support. The authors should also consider and think about possible intervening variables that could have influenced these results. For example, age of the participants and gender.

Answer;

It is a very good idea that we didn’t think of. Regarding this, we added the following to the discussion section.

The second explanatory hypothesis will require further research to support it. However, the reason we deduced this was that the mean age of the study population was in their mid-40s, the most socially active. In addition, we also considered a slight decrease in the proportion of male patients using EMS as MHC after COVID-19 (from 54.3% to 45.7%, P value 0.03).

  • Also, the interpretation considering refusing transportation should be taken with caution, also because the increasing rate was not significant.

Answer;

We agree with your comment. Therefore, we revised the sentence of discussion as follows.

Before

The noteworthy finding in our study was the increased rate of patients refusing transportation even though the finding was not significant (26.49% before COVID-19 versus 28.53% during COVID-19, P = 0.161).

After

What is noteworthy in our study is that we identified patients refusing transportation.

  • In general, results should be interpreted also considering that some patients could not have used EMCs, and, therefore, generalization is not possible.

Answer;

We completely agree with you. We added the following sentence to the limitation section.

First, the EMS system varies by country, and some patients may not be able to use EMS. In some cases of suicide where the cause of death is not clear, administrative procedures may proceed without contact with EMSs. These cases would not have been included in this study. Thus it may not be reasonable to generalize this result.

Round 2

Reviewer 2 Report

Dear Authors, 

I believe the work has benefited from all comments and suggestions. Also, the majority of comments have been addressed satisfactorily. However, the following comment is still not clear to me. 

Considering “study population” the authors reply <<We did not set age exclusion criteria, considering that participants may have a different data pattern>> however, the decision of not considering participants’ age should be better specified and justified. 

Author Response

Thank you for your valuable time and comments.  

Comments and Suggestions for Authors

Dear Authors, 

I believe the work has benefited from all comments and suggestions. Also, the majority of comments have been addressed satisfactorily. However, the following comment is still not clear to me. 

Considering “study population,” the authors reply <<We did not set age exclusion criteria, considering that participants may have a different data pattern>> however, the decision of not considering participants’ age should be better specified and justified. 

→ Answer

Thank you for your valuable time and comment. We admit that we did not clearly describe it.

In collecting data, we did not set an age limit. We did not exclude young age patients.  We want to identify the trend of EMS use in local pediatric MHC patients, which has never been investigated.

However, there were few MHC patients under 18 years ( 95 patients during PreCOVID-19 and 65 patients during COVID-19 periods). Therefore, further analysis was not conducted. The youngest was ten years old in PreCOVID-19 and nine years old in COVID-19. The most senior age was 99 years old in PreCOVID-19 and 98 years old in COVID-19.

We revised the sentences in the Study population section as follows.

Before

The inclusion criteria were all patients who used EMSs for MHCs during the study period. We did not set age exclusion criteria, considering that the participants may have different data patterns.

After

The inclusion criteria were all patients who used EMSs for MHCs during the study period. We did not set exclusion criteria for age to investigate the trend of EMS use in MHC patients under the young age population in the region.

And we added the following texts to the Results section.

There were 359,496 EMS dispatches during the study periods (186,986 before COVID-19 and 172,510 during COVID-19). We excluded 69,866 non-rescue dispatches, such as those involving public support, vehicle maintenance, and education; missing records; and cancellations, and finally analyzed 289,630 dispatches (152,747 before the COVID-19 period and 136,886 in the COVID-19 period). Among them, we extracted 8,577 patients according to the inclusion criteria, 4571 patients before COVID-19, and 4,006 patients during COVID-19 (Figure 1).

The mean age of the PreCOVID-19 group was 46.51 ±18.13, which was slightly older than the COVID-19 group (45.44 ± 18.63), and the proportion of males was 45.44%, which was higher than the COVID-19 group. There were 95 patients under 18 years old in the PreCOVID-19 group and 65 in the COVID-19 group. The youngest was ten years old, the oldest was 99 years old in the PreCOVID-19 group, and nine years old and 98 years of age in the COVID-19 group.

Thank you for your consideration. We look forward to hearing from you.
